# Simulated Adaptive Radiotherapy for Shrinking Glioblastoma Resection Cavities on a Hybrid MRI–Linear Accelerator

**DOI:** 10.3390/cancers15051555

**Published:** 2023-03-02

**Authors:** Beatriz Guevara, Kaylie Cullison, Danilo Maziero, Gregory A. Azzam, Macarena I. De La Fuente, Karen Brown, Alessandro Valderrama, Jessica Meshman, Adrian Breto, John Chetley Ford, Eric A. Mellon

**Affiliations:** 1Department of Radiation Oncology, Sylvester Comprehensive Cancer Center, University of Miami Miller School of Medicine, Miami, FL 33136, USA; 2Department of Biomedical Engineering, University of Miami, Coral Gables, FL 33146, USA; 3Department of Radiation Medicine & Applied Sciences, UC San Diego Health, La Jolla, CA 92093, USA; 4Department of Neurology, Sylvester Comprehensive Cancer Center, University of Miami Miller School of Medicine, Miami, FL 33136, USA

**Keywords:** radiotherapy, glioblastoma, resection cavity, hippocampi, cognitive function, dose reduction

## Abstract

**Simple Summary:**

Cognitive function after brain radiation therapy (RT) is correlated with radiation doses to the normal brain and hippocampi. During the RT of glioblastoma, daily magnetic resonance imaging (MRI) by combination MRI–linear accelerator (MRI–Linac) systems has demonstrated significant anatomic changes due to evolving post-surgical cavity shrinkage. Therefore, this study aimed to investigate if adaptive planning to the shrinking target could reduce the normal brain RT dose with the goal of improving post-RT function. We evaluated a cohort of 10 glioblastoma patients previously treated on a 0.35T MRI–Linac with a prescription of 60 Gy delivered in 30 fractions over six weeks without adaptation (“static plan”) with concurrent temozolomide. Six weekly plans were created per patient. Reductions in radiation dose to the hippocampi (maximum and mean) and brain (mean) were observed for weekly adaptive plans. Weekly adaptive re-planning has the potential to spare the brain and hippocampi from high-dose radiation, likely reducing the neurocognitive side effects of RT.

**Abstract:**

During radiation therapy (RT) of glioblastoma, daily MRI with combination MRI–linear accelerator (MRI–Linac) systems has demonstrated significant anatomic changes, including evolving post-surgical cavity shrinkage. Cognitive function RT for brain tumors is correlated with radiation doses to healthy brain structures, especially the hippocampi. Therefore, this study investigates whether adaptive planning to the shrinking target could reduce normal brain RT dose with the goal of improving post-RT function. We evaluated 10 glioblastoma patients previously treated on a 0.35T MRI–Linac with a prescription of 60 Gy delivered in 30 fractions over six weeks without adaptation (“static plan”) with concurrent temozolomide chemotherapy. Six weekly plans were created per patient. Reductions in the radiation dose to uninvolved hippocampi (maximum and mean) and brain (mean) were observed for weekly adaptive plans. The dose (Gy) to the hippocampi for static vs. weekly adaptive plans were, respectively: max 21 ± 13.7 vs. 15.2 ± 8.2 (*p* = 0.003) and mean 12.5 ± 6.7 vs. 8.4 ± 4.0 (*p* = 0.036). The mean brain dose was 20.6 ± 6.0 for static planning vs. 18.7 ± 6.8 for weekly adaptive planning (*p* = 0.005). Weekly adaptive re-planning has the potential to spare the brain and hippocampi from high-dose radiation, possibly reducing the neurocognitive side effects of RT for eligible patients.

## 1. Introduction

Glioblastoma is the most common primary brain cancer worldwide and is frequently fatal. The standard of care for glioblastoma is maximal safe surgical resection followed by daily radiation and chemotherapy for 6 weeks, and then continued chemotherapy with temozolomide for 6 months [1].

Brain tumors and their treatment (surgery, chemotherapy, and radiation) can affect the patient in multiple ways. For example, cognitive decline can be a delayed side effect of radiation therapy, which can include attention problems, memory, and the speed of processing information. Approximately 30% to 50% of brain tumor patients present with cognitive decline symptoms 6 months after radiation therapy (RT) [2]. Studies suggest that the irradiation of the hippocampus and healthy brain tissue is correlated to post-treatment cognitive dysfunction [3].

Hybrid MRI–linear accelerator (MRI–Linac) systems have gained in popularity since their initial 510(k) marketing authorization in 2012 [4]. Such systems offer improved soft tissue delineation over cone beam computed tomography (CBCT)-guided RT [5,6]. A previous study has shown the possibility of the daily tracking of glioblastoma resection cavity (RC) volumes during RT [7]. RC volumes play a significant role in the treatment planning process, as RT guidelines for glioblastoma include the treatment of the RC plus additional margins [7,8]. Changes to the RC volume have consequences for the RT dose distribution in structures surrounding the RC. The standard of care for glioblastoma RT is based on one plan after the surgical resection of the tumor, and is usually generated about a week or more prior to the RT start date. However, the daily MRI–Linac treatments have permitted the visualization of significant anatomical changes during RT, including evolving RC shrinkage [9,10]. Here, we demonstrate that RC shrinkage can be adapted on the MRI–Linac platform with savings in seemingly normal brain structure doses.

## 2. Materials and Methods

### 2.1. Daily Set up Images

Daily set-up images were acquired before treatment delivery on a 0.35-T MRI–Linac (ViewRay MRIdian, Cleveland, OH, USA). Patients were immobilized in a custom thermoplastic mask, and they were imaged with a vendor-supplied head and neck anterior flexible coil and torso posterior flexible coil wrapped around the thermoplastic mask and baseplate (total 11 channels). Images were acquired using a balanced steady-state free procession pulse sequence (bSSFP), characterized by its short acquisition time and high signal-to-noise ratio. Images acquired with this pulse sequence combine T1/T2 contrast behavior [11]; however, at the high flip angles used here the images are predominantly T2-weighted as we have previously demonstrated [7]. The set-up MRI scans used for this study were acquired with 1.5 × 1.5 × 1.5 mm voxel dimension, TR/TE = 3.35/1.45 ms, flip angle = 60 degrees and bandwidth = 536 Hz/pixel. The field-of-view for the set-up scans was chosen to fit the entire head and the acquisition time was 172 s.

### 2.2. Patient Data Base and Analysis

Ten patients were selected from a prospective non-interventional study of 0.35-T MRI–Linac glioblastoma RT. Thirty-six patients were evaluated for possible inclusion in this study. Of the 36 patients analyzed, 26 had a gross total or near-total resection defined as resection cavity only on MRI (no enhancing tumor lesion) or resection cavity with small tumor lesion and/or a postoperative change of less than 20 mL. Near total resection or gross total resection was defined by an attending radiation oncologist after evaluating the surgery note from the neurosurgeon, the radiology notes from the radiologist, and the post-surgery imaging. Of those 26 patients, 13 had cavity shrinkage during RT, defined as any volume decrease with a volume reduction range from the planning of the MRI RC to the beginning of Week 5 of treatment MRI RC of: −0.88 mL to −12.47 mL. Of those 13 patients, 3 were excluded from analysis due to the location of the tumor involving both hippocampi. Daily T2-weighted TRUFI bSSFP treatment set-up MRI scans from the MRI–Linac were transferred to radiation oncology software (MIM Software, Cleveland, OH, USA) where RC, planning tumor volume (2 cm clinical target volume expansion within brain followed by 3 mm expansion of RC), brain, and organs at risk (OARs) were contoured on the planning scan and last day of every treatment week (Fractions 5, 10, 15, 20, 25). These contours were reviewed by a senior radiation oncologist. Figure 1 shows the structures that were contoured for most patients. Constraints for brainstem, optic structures, eyes, and lenses are typically included on modern cooperative group protocols for brain radiotherapy, as excess dose can cause vision loss, cranial neuropathies, or paralysis [11,12,13]. We also include hippocampi, as emerging data suggest that hippocampal doses are correlated to neurocognitive function after radiotherapy [14], and lacrimal glands, as excess doses can cause permanent xeropthalmia [15].

Fractionated RT plans were created for every patient, with a prescription of 60 Gy to the planning target volume (PTV). The PTV is the sum of the clinical target volume (CTV), a 2 cm expansion from the resection cavity within the normal brain confined by anatomic barriers such as dura, plus 3 mm for daily setup error. Treatment was delivered over a period of six weeks (5 days per week, 2 Gy per day). These plans that were not adapted are denoted as static plans. Diagnostic and MRI–Linac MRIs and computed tomography (CT) scans taken pre-RT were used to plan the individual static plans for the MRI–Linac planning system (Figure 2A). The dose distribution to organs at risk and the brain were recorded.

For the adaptive plans, MRI–Linac set-up scans from fractions 5, 10, 15, 20, and 25 (the last days of each week of treatment) were used to calculate treatment plans for the following five fractions of treatment. Contours were created for each fraction to account for changes in RCs and shifts in OARs. For consistency, constraints on each weekly plan were kept constant (including the threshold and importance values). Each plan was normalized to 95% coverage at prescription dose to the PTV. The weekly plans were divided in dosage, prescribing 10 Gy per week for six weeks, summing a total of 60 Gy (Figure 2B).

The adaptive plans were then transferred to MIM software. After fusing all plans to the pre-RT scan as a reference, a summation of dose was performed. The accumulated dose was then recorded for OARs and healthy brain. Dose values to the hippocampus recorded from non-adaptive plans and adaptive plans were compared using a paired *t*-test, and the statistical analysis was conducted in Minitab. We considered *p* values < 0.05 to be statistically significant.

## 3. Results

The mean and max doses for the static and adaptive plans for the uninvolved hippocampi and brain (Table 1) and a summary of statistical values (Table 2) are shown. For every patient, there is a dose reduction to the hippocampus and brain tissue, with dose to healthy tissue trending downwards (Figure 3A) and isodose lines freeing the hippocampi (Figure 3B) as the weeks progressed (Table 1). Only in patients 3 and 5 did the dose to OARs not change significantly during treatment due to cavity position superior in the brain, far from the hippocampi. The difference between the hippocampus mean (*p* = 0.003) and max (*p* = 0.036) and brain mean (*p* = 0.005) for the static and adaptive plans all reached statistical significance (Figure 4, Table 2). From the 10 patients evaluated, 9 also showed dose reductions to optic structures (mean static plan 96.7 Gy, mean adaptive plan 84.5 Gy, *p* = 0.044) and brainstem (mean static plan 18.21, mean adaptive plan 14.5, *p* = 0.041).

## 4. Discussion

MRI-guided RT can reduce doses to healthy tissue by adapting treatment plans to changes during RT [16]. Brain tumor resection cavities are clearly visible on hybrid MRI–RT systems without exogenous contrast, and the cavities of a significant percentage of patients with gross or near total resection (36% of patients analyzed) shrink during treatment. This study demonstrates that weekly adaptive re-planning in glioblastoma can lower the dose to healthy brain tissue by adapting to shrinking resection cavities.

Adaptive RT might improve glioblastoma RT toxicity as hippocampal RT doses have been correlated with worsened neurocognitive outcomes, while sparing the hippocampi from irradiation has been correlated with improved neurocognitive outcomes [14]. Studies have shown that increasing the hippocampal dose correlates with declines in neurocognitive outcomes [17], and sparing the hippocampi from doses above 55 Gy [18] improves neurocognitive outcomes. Similarly, for normal-appearing uninvolved brain, radiation doses beyond 35 Gy cause cortical atrophy on serial imaging that may lead to decreased neurocognition [19]. Furthermore, doses as low as 10 Gy disrupt white matter connectivity [20]. Another technique that limits brain RT dose is proton therapy [21]. A randomized trial of protons vs. photons in glioblastoma for improved brain sparing reduced grade 2 toxicities and high-grade lymphopenia [22,23], which might also be possible with other brain-sparing techniques such as this one. Adaptation as demonstrated here should not negatively impact tumor control, as tumor cells would be expected to shift with the normal-appearing brain towards the shrinking resection cavity.

As an alternative to hybrid MRI–RT systems, standalone MRI during RT has been proposed to reduce boost margins during RT to reduce the volume of brain treated [24] in efforts to limit toxicity. This has been shown to improve target coverage and reduce the dose to the normal brain [25]. One study has even suggested that such an adaptive strategy could improve patient overall survival [26]. The current work expands on these existing studies to demonstrate feasibility on an integrated MR–Linac system with integrated adaptive radiotherapy capabilities, and demonstrates improvement in hippocampal sparing, which could improve patient quality of life.

The main limitation for the integration of standalone MRI during RT in clinical practice is the cost, availability, and coordination of standalone MRIs for patients during RT as well as the extra effort for re-simulation and planning. Hybrid MRI–RT systems essentially nullify these concerns since they allow for treatment and MRI in the same session and include an adaptive online radiotherapy workflow [27]. To this end, the UNITED trial (NCT04726397) uses a hybrid MRI–RT system with weekly adaptation to intact disease and cavity based on T1 post-contrast MRI with 5 mm CTV margins. In this study, we propose weekly adaptation on non-contrast T2-weighted or FLAIR images, as the US Food and Drug Administration has advised to “minimize repeated [gadolinium contrast] imaging studies when possible, particularly closely spaced MRI studies [28]”. In the case of rapid tumor recurrence during RT, an expansion of edema on T2 or FLAIR would be visible around the resection cavity, and this could trigger gadolinium contrast administration in select patients.

## 5. Conclusions

The weekly adaptive MRI-guided re-planning of shrinking glioblastoma resection cavities has the potential to spare brain and hippocampi from high-dose radiation, reducing RT toxicity. Clinical trials of MRI-guided adaptive RT should measure its impact on neurocognition and other toxicities.

## Figures and Tables

**Figure 1 cancers-15-01555-f001:**
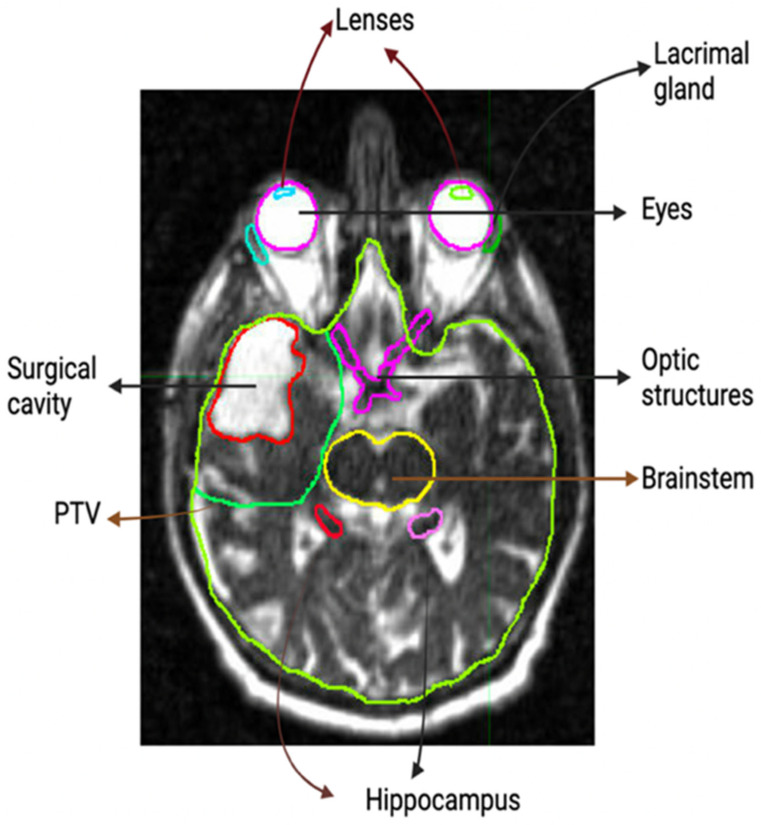
Contoured structures on the MRI scan: RC (red), PTV (green), Brainstem (yellow), right hippocampus (red), left hippocampus (pink), optic nerves (magenta), eyes (magenta), lenses (blue, green), lacrimal glands (blue, green).

**Figure 2 cancers-15-01555-f002:**
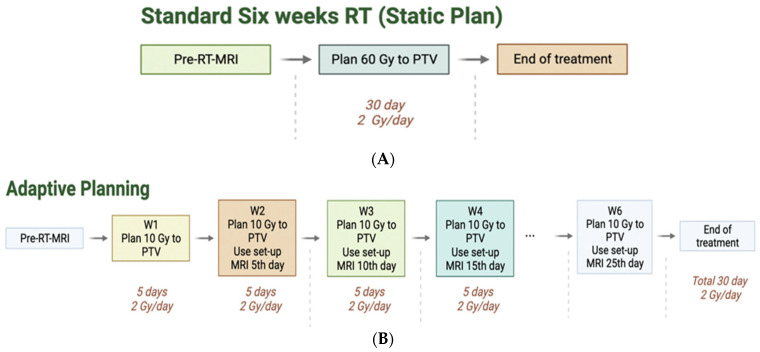
(**A**): Flow chart for static plan (60 Gy prescription to the PTV). (**B**): Flow chart for Adaptive planning (10 Gy prescription to the PTV with a total of 6 different plans and accumulated dose of 60 Gy). W1–W6 stands for Week 1–Week 6 of RT.

**Figure 3 cancers-15-01555-f003:**
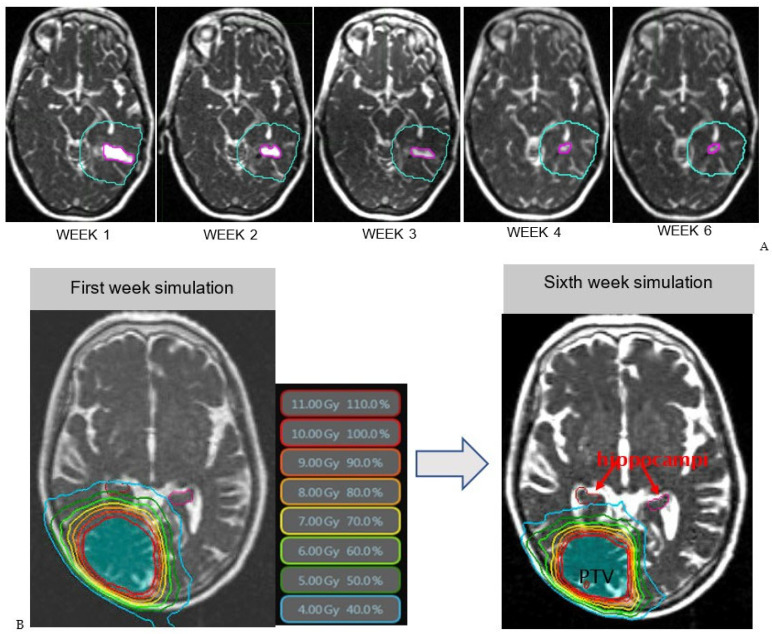
(**A**): Resection cavity shrinkage visualized on MRI–Linac: T2-weighted bSSFP treatment set-up scans This shows the PTV and RC volume trending downwards as the weeks progressed. Week 1: Volume of RC is 3.0 mL and mean dose to brain is 3.3 Gy. Week 2: Volume of RC is 1.3 mL and dose to brain is 2.7 Gy. Week 3: Volume of RC is 0.8 mL and dose to brain is 2.7 Gy. Week 4: Volume of RC is 0.6 mL and dose to brain is 2.5 Gy. Week 6: Volume of RC is 0.5 mL and dose to brain is 2.5 Gy. (**B**): Example adaptive replanning comparing the dose distribution for the first week of treatment (left) and the dose distribution for the sixth week (right). Isodose lines are shown in color surrounding the PTV (area shaded in blue). Both left and right hippocampi are contoured, shown in pink (left) and red (right). During the first week of treatment, the 4 Gy isodose line (light blue) completely covers the right hippocampus. Comparing the 4 Gy isodose line between the two scans, at Week 6, this line has reduced radius sparing the right hippocampus from radiation dose. The dose also decreases to healthy tissue surrounding the PTV outside of the hippocampi.

**Figure 4 cancers-15-01555-f004:**
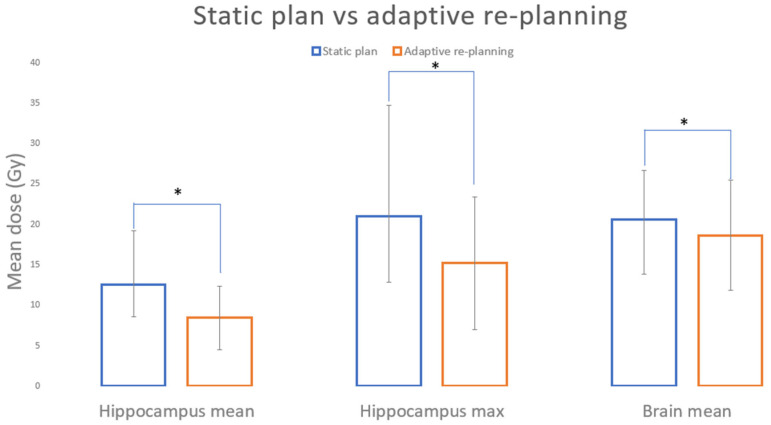
Static plan vs. adaptive re-planning (bars represent standard deviation). * Indicates statistical significance between the static and adaptive plans.

**Table 1 cancers-15-01555-t001:** Static vs. adaptive plans for all patients. Mean and maximum (max) dose to hippocampus and brain. All numbers shown are in units of Gy.

PATIENT #	SUM HIPPOCAMPUS				BRAIN		UNINVOLVED HIPPOCAMPUS
	Static Plan		Adaptive Plan		Static Plan	Adaptive Plan	
	mean	max	mean	max	mean	mean	
1	23.8	44.4	11.8	19.9	23.0	18.5	both
2	12.2	16.8	8.9	11.2	14.3	9.7	left
3	2.4	3.7	2.0	2.9	13.0	11.2	both
4	11.0	13.7	7.6	12.1	26.4	26.0	left
5	4.6	9.6	3.9	6.9	19.5	17.9	both
6	10.4	17.0	7.2	15.6	19.4	15.9	right
7	22.8	29.2	16.4	21.5	22.7	22.4	both
8	13.1	43.6	9.5	32.4	21.3	19.2	both
9	12.3	16.2	8.7	15.1	14.3	13.7	both
10	12.9	15.9	8.4	14.5	32.5	32.1	right

**Table 2 cancers-15-01555-t002:** Statistical data for static and adaptive plans. All numbers shown are in units of Gy. Sum hippocampus refers to the accumulated dose between the left and right hippocampi (involved hippocampi excluded).

	STATIC PLAN (SP)	ADAPTIVE RE-PLANNING	STDEV SP	STDEV AP	*p*-VALUES
Hippocampus mean	12.5	8.4	6.7	3.96	0.003
Hippocampus max	21	15.2	13.7	8.19	0.036
Brain mean	20.6	18.7	6.0	6.81	0.005

## Data Availability

The data presented in this study will be submitted to The Cancer Imaging Archive pending acceptance of this manuscript.

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
