# Peer review of "Simulated Adaptive Radiotherapy for Shrinking Glioblastoma Resection Cavities on a Hybrid MRI–Linear Accelerator"

_cancers, 2023, doi:10.3390/cancers15051555_

Round 1
Reviewer 1 Report
The manuscript addresses a relevant topic: adaptive radiotherapy for shrinking glioblastoma resection cavities on a hybrid MRI-Linear Accelerator.
The authors overserve that for the hippocampus mean and max dose as well as the mean brain dose are significantly lower in adaptive replanning. They conclude that weekly adaptive re-planning has the potential to spare the brain and hippocampi from high-dose radiation, possibly reducing the neurocognitive side effects of RT for eligible patients.
This is a well written manuscript describing a relevant topic that deserves to be published. I have no further objections.
Author Response
Response to Reviewers
(Note: Reviewers text is in plain face, our responses are underlined)
This is an interesting submission by one of the guest editors to their special issue on Advanced Imaging in Brain Tumor Patient Management. Overall, the reviewers had favorable comments, although a few issues require careful attention by the authors and corresponding extensive revisions.
Thank you for your time and thoughtful critique of our manuscript. We believe we have made all requested updates in hopes that the work will now be suitable for publication. If there is anything we missed or any further issues, please let us know.
The manuscript is very specialized and clearly written by radiation oncologists for a radiation oncologist readership (and targeted for inclusion in a special issue on brain tumor imaging). Nonetheless, it would be productive to consider a broader readership and add further explanatory details here and there. For example, in the context of Figure 1, all that is mentioned in the Results section is that it “shows the structures that were contoured for most patients”. While this statement is correct, it would be informative to add further descriptions and explanations as to the relevance of the structures, and which ones are the most important to contour.
Further description has been added, explaining the importance and relevance of countering organs at risk.
It would also be helpful to define all acronyms, in particular the term PTV, which is central to this report. CT needs a definition, too, and so does MRI.
Thank you, these have been added.
Content and conclusions of this report do not appear to add major new insights when considered within the background of studies published by others, although confirmatory results may be considered of interest, too. Some earlier studies are cited, but some are omitted and should be mentioned and cited, such as Vegvary et al., 2020 (doi: 10.21873/anticanres.14425) and Matsuyama et al., 2022 (doi: 10.1186/s13014-022-02007-4).
Thank you for pointing out these interesting studies. We think that this manuscript presents a significant insight into how the workflow will look for online adaptive radiotherapy on a MRI-Linac system. The studies of Vegvary and Matsuyama have been added to the discussion, as they do provide further evidence for the importance of adaptive radiotherapy in glioblastoma. However, both studies only look at two time points during RT. We are replanning at six timepoints, which can be performed using integrated MRI-RT workflow, as we plan for future clinical trials.
As a very minor issue in the Acknowledgements: First person singular is used (“I am grateful”…), which leaves it undefined who is grateful.
This has been clarified.
A more important issue is figure quality, which does not meet publication-quality standards. The images in Figures 1 and 3 are neither high resolution nor well labeled and explained. Major revisions are required.
Figure 1: The MRIs are fuzzy. The same colors should be used to contour the same structures in A and B; e.g. brain stem is green in A, but yellow in B. That’s confusing. Colors should be more clearly distinguished: dark green and light green and turquoise look very similar in A. Gold circles are not defined in A. In B, as mentioned above, it is unclear why the eyes, lenses, lacrimal gland, optic structures/nerves are contoured; please provide background and explanations. What is the purpose of showing 1B: does it serve as a general legend to define the structures shown in 1A?
Figure 1A has been deleted, and Figure 1B changed to Figure 1. We feel that Figure 3A is a better representation of reception cavity shrinkage. Figure 1 shows all the organs at risk that were considered when planning each RT case. In the methods we added:
“Figure 1 shows the structures that were contoured for most patients. Constraints for brain-stem, optic structures, eyes, and lenses are typically included on modern cooperative group protocols for brain radiotherapy, as excess dose can cause vision loss, cranial neu-ropathies, or paralysis. We also include hippocampi, as emerging data suggests that hip-pocampal doses are correlated to neurocognitive function after radiotherapy, and lacrimal glands, as excess dose can cause permanent xeropthalmia.”
Regarding fuzziness, Figure 1 has been magnified to show detail. Please keep in mind that the resolution of the 0.35T MRI is 1.5 mm isotropic, which tends not to have the same detail as a high field MP-RAGE (for example). Also, the contrast between white and gray matter is reduced compared to a pure T1 or T2 image because of the bSSFP imaging contrast.
Figure 3: The MRIs are fuzzy and the colored circles are difficult to see. In part A, the 5 MRIs should be enlarged (maybe put part B below part A, then extend part A all across the page) and the colored circles need to be sharper. In B, the inset with color coding is illegible. The red arrows (pointing to the hippocampi?) are not defined; there are green letters, but it is not possible to read them.
Figure 3A have been enlarged and color circles thickness has been augmented. The arrows pointing to the hippocampi have been defined and green text was replaced by bold red text.
While this report considers radiation treatment (RT) toxicity (neurocognition etc.), there is no discussion of therapeutic efficacy. One can reasonably expect that reducing the PTV, based on a shrinking resection cavity, might result in less severe RT toxicity. However, therapeutic efficacy has to be considered as well, and might even receive priority over side effects of RT. GBM is a highly infiltrative disease, and it is likely that peritumoral normal brain tissue harbors disseminated GBM cells that remain after surgery. It should be discussed whether shrinking the PTV would run the risk of not effectively reaching disseminated GBM cells with RT.
This is an interesting point, and well taken in the context of other studies that use CTV margins of 5 mm. The CTV remains 2 cm throughout this study with a 3 mm PTV added. We added a statement to address this concern in the text as “Adaptation as demonstrated here should not negatively impact tumor control, as tumor cells would be expected to shift with the normal appearing brain towards the shrinking resection cavity.”
Reviewer 2 Report
Cognitive function after brain radiation therapy (RT) is correlated with radiation doses to the normal brain and hippocampi. During RT of gliomas (GBM), daily MRI by combination MRI-linear accelerator (MRI-Linac) systems has demonstrated significant anatomic changes due to evolving post-surgical cavity shrinkage. Therefore, this study aimed to investigate if adaptive planning to the shrinking target could reduce normal brain RT dose with the goal of improving post-RT function. They evaluated a cohort of 10 GBM patients previously treated on a 19 0.35T MRI-Linac with a prescription of 60 Gy delivered in 30 fractions over six weeks without adaptation (“static plan”) with concurrent temozolomide. Six weekly plans were created per patient. Reductions in radiation dose to the hippocampi (maximum and mean) and brain (mean) were observed. Weekly adaptive re-planning has the potential to spare the brain and hippocampi from high-dose radiation, likely reducing the neurocognitive side effects of RT. However the effect of this modification of the radiation dose and potential improvement in cognitive function with the overall survival of these patients was not discussed.
Author Response

(The authors gave the same response as above.)

Round 2
Reviewer 2 Report
Cognitive function after brain radiation therapy (RT) is correlated with radiation doses to the normal brain and hippocampi. During RT of gliomas (GBM), daily MRI by combination MRI-linear accelerator (MRI-Linac) systems has demonstrated significant anatomic changes due to evolving post-surgical cavity shrinkage. Therefore, this study aimed to investigate if adaptive planning to the shrinking target could reduce normal brain RT dose with the goal of improving post-RT function. They evaluated a cohort of 10 GBM patients previously treated on a 19 0.35T MRI-Linac with a prescription of 60 Gy delivered in 30 fractions over six weeks without adaptation (“static plan”) with concurrent temozolomide. Six weekly plans were created per patient. Reductions in radiation dose to the hippocampi (maximum and mean) and brain (mean) were observed. Weekly adaptive re-planning has the potential to spare the brain and hippocampi from high-dose radiation, likely reducing the neurocognitive side effects of RT. However the effect of this modification of the radiation dose and potential improvement in cognitive function with the overall survival of these patients was not discussed.
Author Response
- The legend for figure 2A and 2B has been added to the figure legend outside the image.
2. This has been corrected. We apologized for this. The versions of the manuscript got mixed at some point during the review and the one we submitted did not have this added.
3. The second sentence have been finished.